# LLC Resonant Voltage Multiplier-Based Differential Power Processing Converter Using Voltage Divider with Reduced Voltage Stress for Series-Connected Photovoltaic Panels under Partial Shading

**Masatoshi Uno [1],\* [ID], Toru Nakane [2] and Toshiki Shinohara [3]**

[1]   College of Engineering, Ibaraki University, Hitachi 316-8511, Japan
[2]   Nippon Steel, Tokyo 100-8071, Japan; tn2975@gmail.com
[3]   Toyota Motor Corporation, Toyota City 471-8571, Japan; toshiki_shinohara@mail.toyota.co.jp
\*   Correspondence: masatoshi.uno.ee@vc.ibaraki.ac.jp

**Abstract:** Partial shading on photovoltaic (PV) strings consisting of multiple panels connected in series is known to trigger severe issues, such as reduced energy yield and the occurrence of multiple power point maxima. Various kinds of differential power processing (DPP) converters have been proposed and developed to prevent partial shading issues. Voltage stresses of switches and capacitors in conventional DPP converters, however, are prone to soar with the number of panels connected in series, likely resulting in impaired converter performance and increased circuit volume. This paper proposes a DPP converter using an LLC resonant voltage multiplier (VM) with a voltage divider (VD) to reduce voltage stresses of switches and capacitors. The VD can be arbitrarily extended by adding switches and capacitors, and the voltage stresses can be further reduced by extending the VD. Experimental verification tests for four PV panels connected in series were performed emulating partial shading conditions in a laboratory and outdoor. The results demonstrated the proposed DPP converter successfully precluded the negative impacts of partial shading with mitigating the voltage stress issues.

**Keywords:** photovoltaic system; voltage divider; partial shading; differential power processing converter; LLC converter; voltage multiplier

---

## 1. Introduction

Ordinary photovoltaic (PV) panels generally consist of two or three substrings, each comprising multiple cells connected in series. PV panels are further connected in series to meet voltage requirements of loads or systems. When partial shading occurs on the series-connected PV panels due to neighboring buildings, fallen leaves, bird dropping, etc., electrical characteristics of series-connected PV substrings or panels (hereafter call panels unless otherwise noted) are mismatched. Characteristic mismatch conditions are also triggered by uneven aging of panels and uneven irradiance due to curved surfaces of PV panels. Under characteristic mismatch conditions, a string current, $I_{string}$, flows through a bypass diode that is connected in parallel with a shaded panel, as illustrated in Figure 1a. The bypassed panel no longer generates power as its voltage is subzero value, significantly reducing the power generation of the string as a whole. An annual energy yield reportedly deteriorates by 20%–30% due to partial shading on a PV panel [1]. Furthermore, partial shading generates multiple maximum power points (MPPs), including one global and some local MPPs, on a *P–V* characteristic of the partially-shaded string that might hinder and confuse ordinary MPP tracking (MPPT) algorithms—the shaded PV panel might operate at a local MPP, at which the panel generates less power than at the global MPP, as shown in Figure 1b. Although advanced MPPT techniques have been proposed to certainly track the global

MPP even under partial shading conditions [2], a shaded panel is bypassed and cannot contribute to power generation, unavoidably resulting in reduced energy yield.

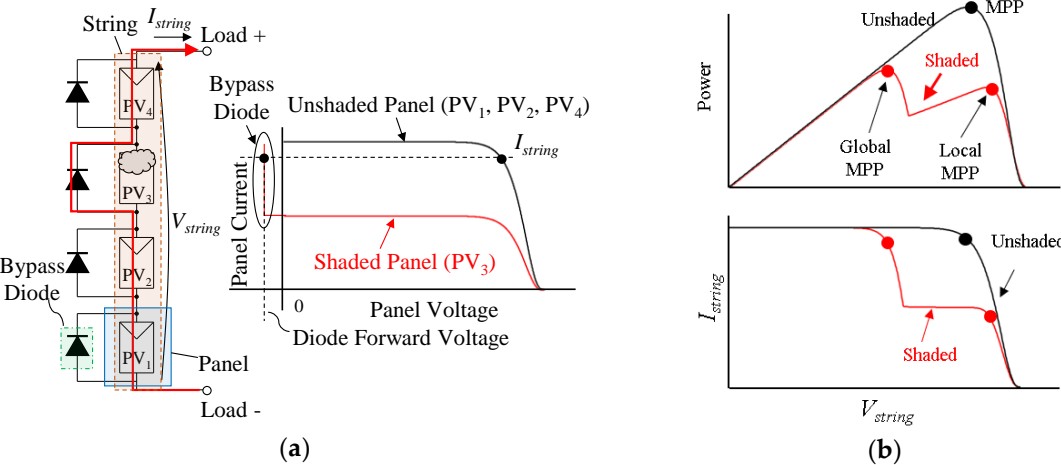

**Figure 1.** Characteristics under partial shading conditions: (**a**) Current path and panel characteristics; (**b**) String characteristics.

Various kinds of differential power processing (DPP) converters, also known as voltage equalizers, have been proposed and developed. DPP converters deliver power from unshaded panels to shaded ones so that all panel characteristics are virtually matched to preclude the abovementioned partial shading issues. DPP converters are roughly categorized into three groups based on power redistribution scenarios, as shown in Figure 2: the adjacent panel-to-panel architecture, panel-to-panel architecture with an isolated port, and string-to-panel architecture. DPP converters in the adjacent panel-to-panel architecture are based on nonisolated bidirectional converters, such as PWM converters [3–10] and switched capacitor converters [11–17]. Isolated bidirectional flyback converters are employed in the panel-to-panel architecture with an isolated port [18–24]. For the string-to-panel architecture, single-input–multi-output converters, such as a multi-winding flyback converter [25], multi-stacked buck-boost converters [26,27], and resonant voltage multipliers (VMs) [28–31], are employed. Regardless of types of architectures, these DPP converters transfer a fraction of the generated power of unshaded panels to shaded ones so that all panel characteristics are virtually matched.

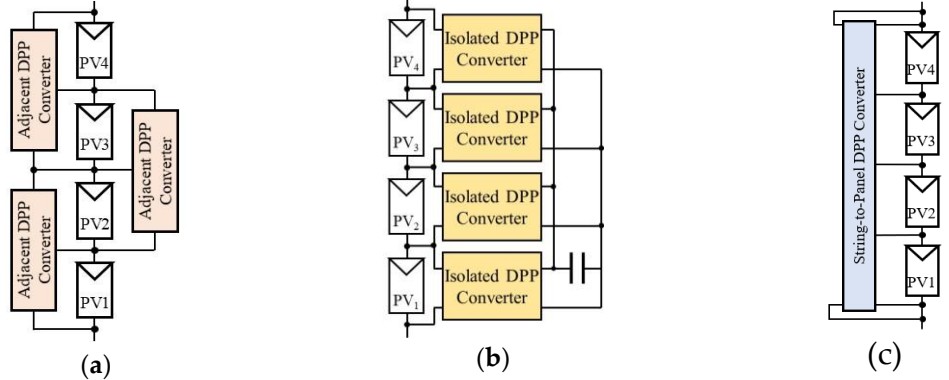

**Figure 2.** Differential power processing (DPP) architectures: (**a**) Adjacent panel-to-panel architecture; (**b**) Panel-to-panel with isolated port architecture; (**c**) String-to-panel architecture.

In PV systems, more than ten panels are connected in series to form a PV string, and an MPP voltage and open-circuit voltage of such strings reach 400–600 V for traditional systems or even 1000 V for the latest high-voltage PV systems. With the adjacent panel-to-panel architecture, shown in Figure 2a, numerous DPP converters in proportion to the number of series-connected panels necessary,

and therefore the total volume, cost, and complexity of the system as a whole are prone to increase. Furthermore, since power transfer is limited only between adjacent two panels, power from unshaded panels might have to traverse multiple DPP converters and panels before reaching shaded panels, hence collectively increasing power conversion losses. With the panel-to-panel architecture with an isolated port, shown in Figure 2b, power from unshaded panels can directly be transferred to shaded ones via the isolated port. However, each DPP converter contains a bulky and expensive transformer, and the number of DPP converters necessary is equal to the number of panels connected in series, thus resulting in increased system complexity and cost. Meanwhile, the number of DPP converters can be ideally reduced to only one in the string-to-panel architecture, as shown in Figure 2c. Furthermore, shaded panels can directly receive power from the string, and hence the overall power conversion efficiency can be improved in comparison with the adjacent panel-to-panel architecture.

Conventional DPP converters have been chiefly developed for substring-level equalization within individual panels, and therefore voltage stresses of semiconductor devices and passive components are rated for panel voltages around 30–50 V. For panel-level equalization, on the other hand, circuit components are exposed to a string voltage $V_{string}$ of several hundred volts. Three-level converters based on a neutral-point diode-clamped converter [32] and an LLC resonant converter (see Figure 3) [33] have been proposed to mitigate voltage stress issues, and voltage stresses of switches and capacitors can be reduced to $V_{string}/2$. However, these circuit components are still exposed to high voltage stresses in high-voltage PV strings.

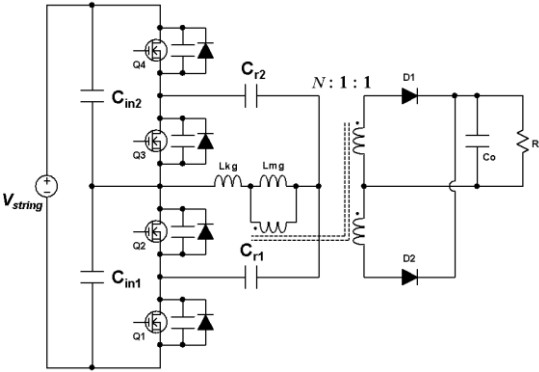

**Figure 3.** Three-level LLC resonant converter.

This article presents a novel DPP converter for the string-to-panel DPP architecture for high-voltage strings. The proposed DPP converter is based on the combination of a conventional LLC resonant VM and voltage divider (VD) using switched capacitor converters (SCCs). In comparison with the conventional DPP converter based on the LLC resonant VM, voltage stresses of switches can be reduced. In addition, voltage stresses of input smoothing capacitors can also be reduced by adding inductors to the SCCs. The rest of this paper is organized as follows. Section 2 discusses the proposed DPP converter and its major features. The operation analysis will be performed in Section 3. Section 4 presents the results of experimental verification tests using a prototype for four or eight panels connected in series in a laboratory and outdoor. Section 5 contains the concluding statements.

## 2. Proposed DPP Converter Based on Voltage Divider and LLC Resonant Voltage Multiplier

### 2.1. Key Circuits for Proposed DPP Converter

From three key circuits of an SCC, bidirectional PWM converter, and conventional LLC resonant VM (see Figure 4), the proposed DPP converter for the string-to-panel architecture can be derived. Switches in the SCC (see Figure 4a) are driven with a fixed 50% duty cycle, and all capacitors in the SCC are indirectly connected in parallel, and their voltages are automatically balanced. With the

bidirectional PWM converter operating with a fixed 50% duty cycle (see Figure 4b), the voltages of the input and output capacitors, $C_{in}$ and $C_{out}$, can be automatically balanced.

The LLC resonant VM, shown in Figure 4c, comprises an LLC resonant inverter and VM. The high- and low-side switches ($Q_b$ and $Q_a$) are driven at a fixed frequency with a fixed 50% duty cycle in a complementary manner, and the LLC resonant tank is driven by a square-wave voltage with a peak-to-peak voltage of $V_{string}$, as shown in the inset of Figure 4c. A sinusoidal current generated by the LLC resonant tank is transferred to the secondary side and is rectified by the VM. Eventually, the VM produces uniform multiple output voltages, by which PV substring characteristics are automatically unified even without feedback control [28–31].

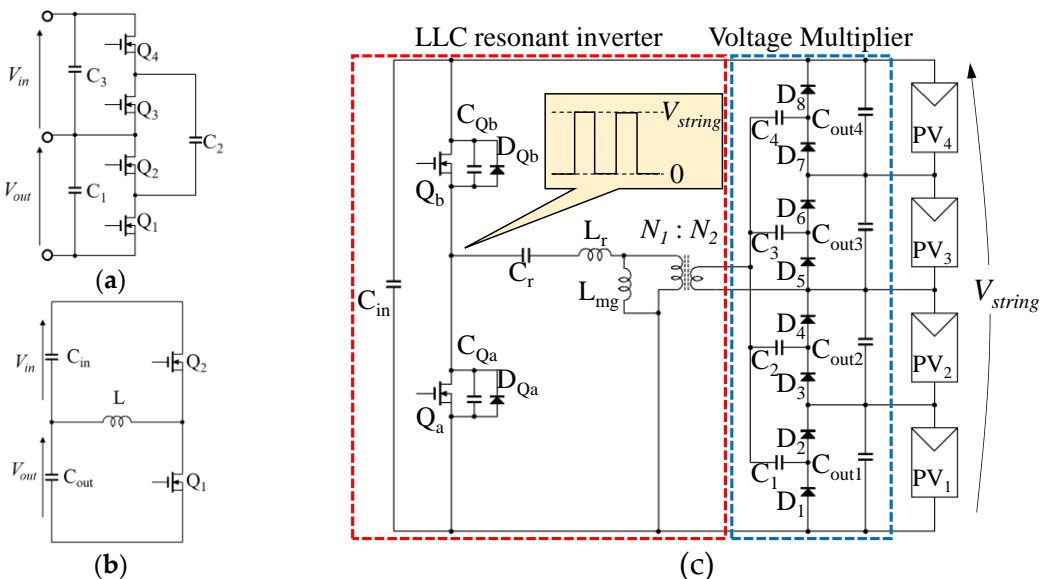

**Figure 4.** Key circuits for proposed DPP converter: (**a**) Switched capacitor converter (SCC); (**b**) Bidirectional PWM converter; (**c**) LLC resonant voltage multiplier (VM).

### 2.2. Proposed DPP Converter

By merging the three circuits shown in Figure 4, the proposed DPP converter for the string-to-panel architecture can be derived, as shown in Figure 5a, in which a topology for four panels connected in series is exemplified. In comparison with the conventional DPP converter shown in Figure 4c, the VM is identical, whereas the LLC resonant inverter side is modified to be a VD with an LLC resonant inverter to reduce voltage stresses of switches $Q_1$–$Q_4$ and input smoothing capacitors $C_{in1}$–$C_{in4}$. Body diodes ($D_{Q1}$–$D_{Q4}$) and parasitic output capacitances ($C_{Q1}$–$C_{Q4}$) of MOSFETs, which play important roles to achieve zero voltage switching (ZVS) operations, are also illustrated. A leakage inductance $L_r$ and magnetizing inductance $L_{mg}$ of the transformer are utilized for the resonant operation. The VD comprises the SCC containing inductors, while the LLC resonant inverter is very similar to that in the conventional DPP converter shown in Figure 4c. For the sake of clarity, the circuit on the VD side is defined as a transformer's primary side, and the secondary side is the VM. The string voltage $V_{string}$ is divided into four by the SCC and PWM converters in the VD, achieving the reduced voltage stresses of circuit components on the primary side.

The VD circuit is emphasized in Figure 5b. Series-connected capacitors (i.e., $C_{in1}$–$C_{in2}$, $C_{in3}$–$C_{in4}$, and $C_{r1}$–$C_{r2}$) act as capacitors $C_1$, $C_2$, and $C_3$ in the traditional SCC (see Figure 4a), respectively. $C_{r1}$ and $C_{r2}$ also behave like resonant capacitors. Meanwhile, thanks to the added inductors of $L_1$ and $L_2$, and the magnetizing inductor $L_{mg}$ of the transformer, three PWM converters can be formed in the VD. By driving switches with a fixed 50% duty cycle, all capacitor voltages are automatically balanced to be $V_{string}/4$ by the SCC and PWM converters in the VD.

The LLC resonant VM is driven by the VD. Resonant capacitors, $C_{r1}$ and $C_{r2}$, and $L_{mg}$ form an LLC resonant tank that produces ac current and voltage. The ac power generated by the LLC resonant tank is transferred to the secondary side, and the transferred power is automatically redistributed to shaded panels having weak characteristics. This power redistribution realizes automatic characteristic equalization. The detailed operation principle and mechanism of this power redistribution have been reported and discussed in detail in the past work [28].

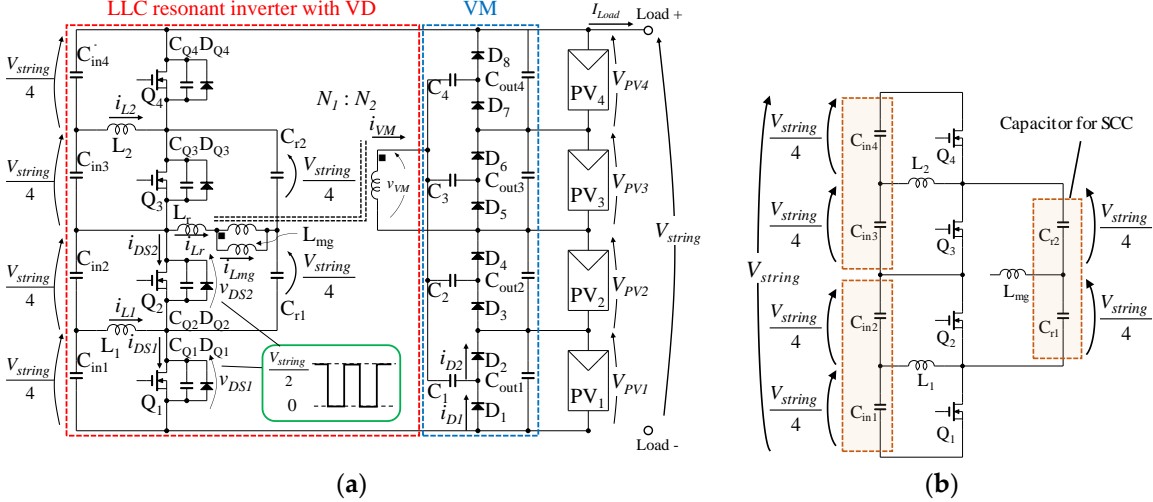

**Figure 5.** (**a**) Proposed DPP converter based on voltage divider (VD) and LLC resonant voltage multiplier (VM); (**b**) Voltage divider.

### 2.3. Major Features

Similar to the conventional LLC resonant VM, a fraction of the generated power of the string is automatically transferred to the shaded panel(s) through the proposed DPP converter even without feedback control. In other words, the proposed DPP converter can operate with a fixed duty cycle at a fixed switching frequency, and no voltage and current sensing are necessary. This automatic power redistribution allows eliminating a feedback control loop from the DPP converter, achieving simplified circuit implementation.

The voltage stresses of switches and capacitors on the primary side can be reduced in comparison with the conventional LLC resonant VM (see Figure 4c). The reduced voltage stresses of capacitors allow small multi-layer ceramic capacitors (MLCCs) to be used even for high-voltage applications. Although multiple inductors are necessary for the VD, these inductors can be very tiny as their average currents are zero.

Another feature of the proposed DPP converter is that the VD is scalable. The VD in the proposed DPP converter can be arbitrarily extended by adding capacitors, switches, and inductors. The larger the number of circuit components in the VD, the lower will be the voltage stresses of capacitors and switches. Figure 6 illustrates the scalability of the VD. The left-hand side circuit consists of four input capacitors connected in series (defined as $n = 4$), and voltage stresses of capacitors and switches are $V_{string}/4$ and $V_{string}/2$, respectively. The circuit on the right-hand side corresponds to the case of $n = 6$, and voltage stresses of capacitors and switches can be reduced to $V_{string}/6$ and $V_{string}/3$, respectively. Either $L_2$ or $L_4$, or both of them, is replaced with a magnetizing inductance of a transformer to form an LLC resonant VM. Similarly, by extending the circuit to $n = 8$, the voltage stresses of capacitors and switches can be further reduced to $V_{string}/8$ and $V_{string}/4$.

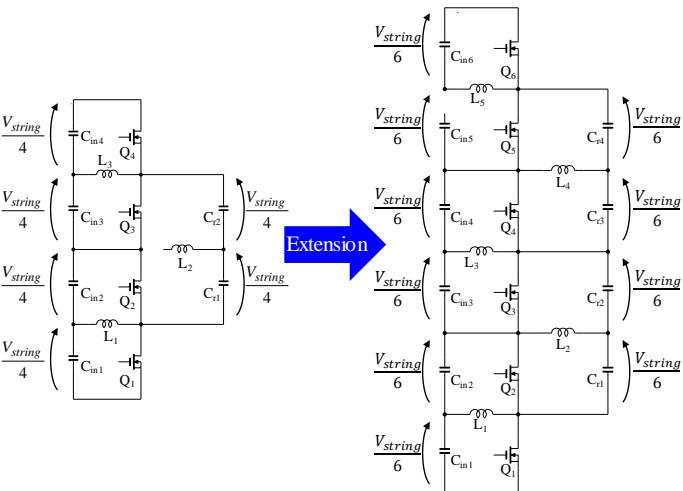

**Figure 6.** Extension of VD from $n = 4$ to $n = 6$.

### 2.4. Extended Topology of Proposed DPP Converter

An extended version of the proposed DPP converter is exemplified in Figure 7. This topology is the combination of a VD with $n = 6$ (see Figure 6) and two LLC resonant VMs (VM$_1$ for PV$_1$–PV$_4$, and VM$_2$ for PV$_5$–PV$_8$). $V_{string}$ is divided into six in the VD, allowing reduced voltage stresses of capacitors and switches, as mentioned in Section 2.3. L$_2$ and L$_4$ in the VD in Figure 6 are replaced with transformers' magnetizing inductances, L$_{mg1}$ and L$_{mg2}$, respectively. An operation mechanism of VMs in the extended topology is identical to that of the VM in Figure 5a. Hence, in the next section, the operation analysis is performed only for the basic topology for the sake of clarity.

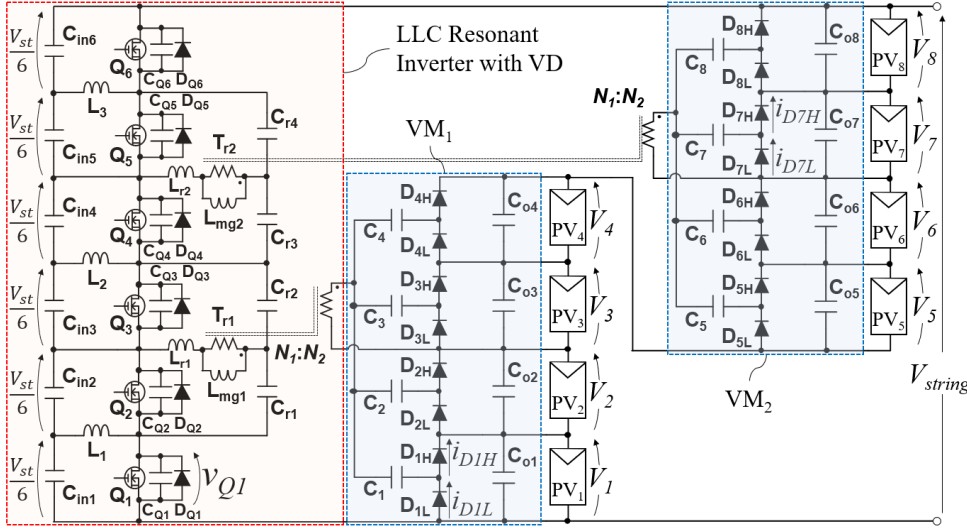

**Figure 7.** Extended DPP Converter using two VMs and VD with $n = 6$.

The increased design difficulty should be cited as a drawback of the proposed DPP converter. The conventional DPP converters for the adjacent panel-to-panel architecture (Figure 2a) and panel-to-panel with an isolated port architecture (Figure 2b) are fully modular and can be applied to any numbers of panels connected in series without design changes. The proposed DPP converter, on the other hand, needs to be redesigned by properly designing the transformers, VD, and VM as the number of panels connected in series changes.

## 3. Operation Analysis

### 3.1. Operation Principle

The operation analysis is performed for the case that $PV_1$ only is partially shaded. Key operation waveforms and current flow paths are shown in Figures 8 and 9, respectively. All switches are driven with a fixed duty cycle with a proper dead time period. Current and voltage waveforms of $Q_1$ and $Q_2$ are identical to those of $Q_4$ and $Q_3$, and therefore, waveforms of $Q_1$ and $Q_2$ only are illustrated and discussed in this section. Capacitances of the resonant capacitors of $C_{r1}$ and $C_{r2}$ are assumed identical.

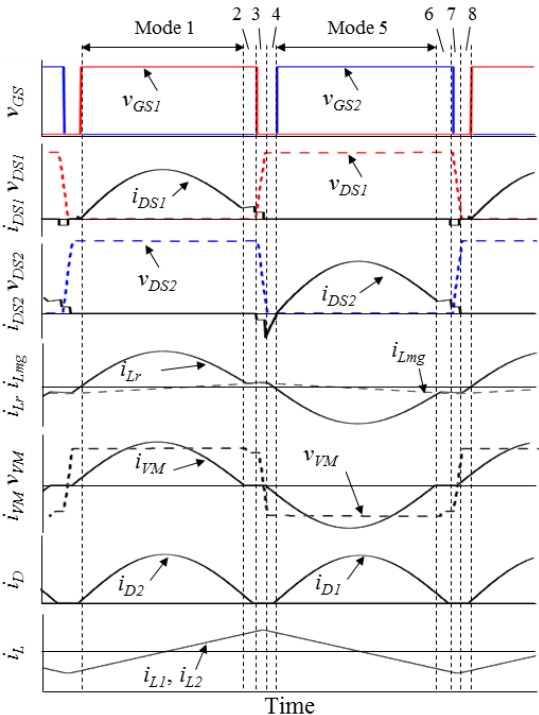

**Figure 8.** Theoretical key operation waveforms when $PV_1$ is shaded.

Mode 1 (Figure 9a): Odd-numbered switches are on, and voltages of $Q_1$ and $Q_2$, $v_{DS1}$ and $v_{DS2}$, are zero and $V_{string}/2$, respectively. The series connection of $C_{in1}$ and $C_{in2}$ is connected in parallel with the series connection of $C_{r1}$ and $C_{r2}$, and therefore, voltages of these series-connected capacitors become uniform. Currents of $L_1$, $L_2$, and $L_{mg}$ ($i_{L1}$, $i_{L2}$, and $i_{Lmg}$) linearly increase as they are charged by $C_{in1}$, $C_{in3}$, and $C_{r2}$, respectively. Meanwhile, the current of $L_r$, $i_{Lr}$, sinusoidally changes due to the resonance between $L_r$ and $C_{r1}$–$C_{r2}$. A current equal to $i_{Lr} - i_{Lmg}$ is transferred to the secondary side in the form of $i_{VM} = N (i_{Lr} - i_{Lmg})$ where $N (= N_1/N_2)$ is the transformer turn ratio. $C_1$ in the VM is charged by $i_{VM}$ through the diode $D_2$. After half the resonant period, $i_{Lr}$ becomes equal to $i_{Lmg}$, and the operation shifts to the next mode.

Mode 2 (Figure 9b): $i_{Lmg}$ and $i_{Lr}$ are equal and linearly increase. In the VD, $i_{L1}$ and $i_{L2}$ still linearly increase. On the secondary side, no currents flow in the VM, except for output smoothing capacitors.

Mode 3 (Figure 9c): The gating signal for $Q_1$, $v_{GS1}$ is removed to turn off $Q_1$, and $C_{Q1}$ and $C_{Q2}$ start to be charged and discharged by $i_{Lmg}$, respectively. $v_{DS1}$ increases with a slope, thus achieving ZVS turn-off. Meanwhile, $v_{DS2}$ decreases with a slope equal to that of $v_{DS1}$ because the sum of $v_{DS1}$ and $v_{DS2}$ is always equal to the total voltage of $C_{in1}$ and $C_{in2}$ (i.e., $V_{string}/2$). This turn-off operation is essentially identical to that of traditional LLC resonant converters.

Mode 4 (Figure 9d): Charging and discharging of $C_{Q1}$ and $C_{Q2}$ are completed. $v_{DS1}$ and $v_{DS2}$ become $V_{string}/2$ and 0, respectively, and the body diode of $Q_2$, $D_{Q2}$, starts conducting. Meanwhile, $i_{L1}$, $i_{L2}$, and $i_{Lmg}$ start linearly decreasing as they are charged by $C_{in2}$, $C_{in4}$, and $C_{r1}$, respectively.

Sinusoidal resonant current $i_{Lr}$ starts flowing again due to the resonance between $L_r$ and $C_{r1}$–$C_{r2}$. In the VM, $i_{VM}$ flows through the low-side diode, $D_1$.

Mode 5: While $D_{Q2}$ is conducting (i.e., $v_{DS2} = 0$), the gating signal for $Q_2$, $v_{GS2}$, is applied to turn on $Q_2$ to achieve ZVS turn-on. This operation mode is symmetric to Mode 1.

Operations in Mode 5–8 are symmetric to those in Mode 1–4 and therefore are omitted to save page length. Since average currents of $i_{L1}$ and $i_{L2}$ are zero, small inductors with a small current rating can be used for $L_1$ and $L_2$. In the VM, currents flow through diodes and a capacitor that are connected to the shaded panel of $PV_1$, while other diodes and capacitors are not in operation. Average current of a capacitor under steady-state conditions must be zero, and therefore, an equalization current supplied from the DPP converter to $PV_1$ is equal to an average current of $D_1$ or $D_2$.

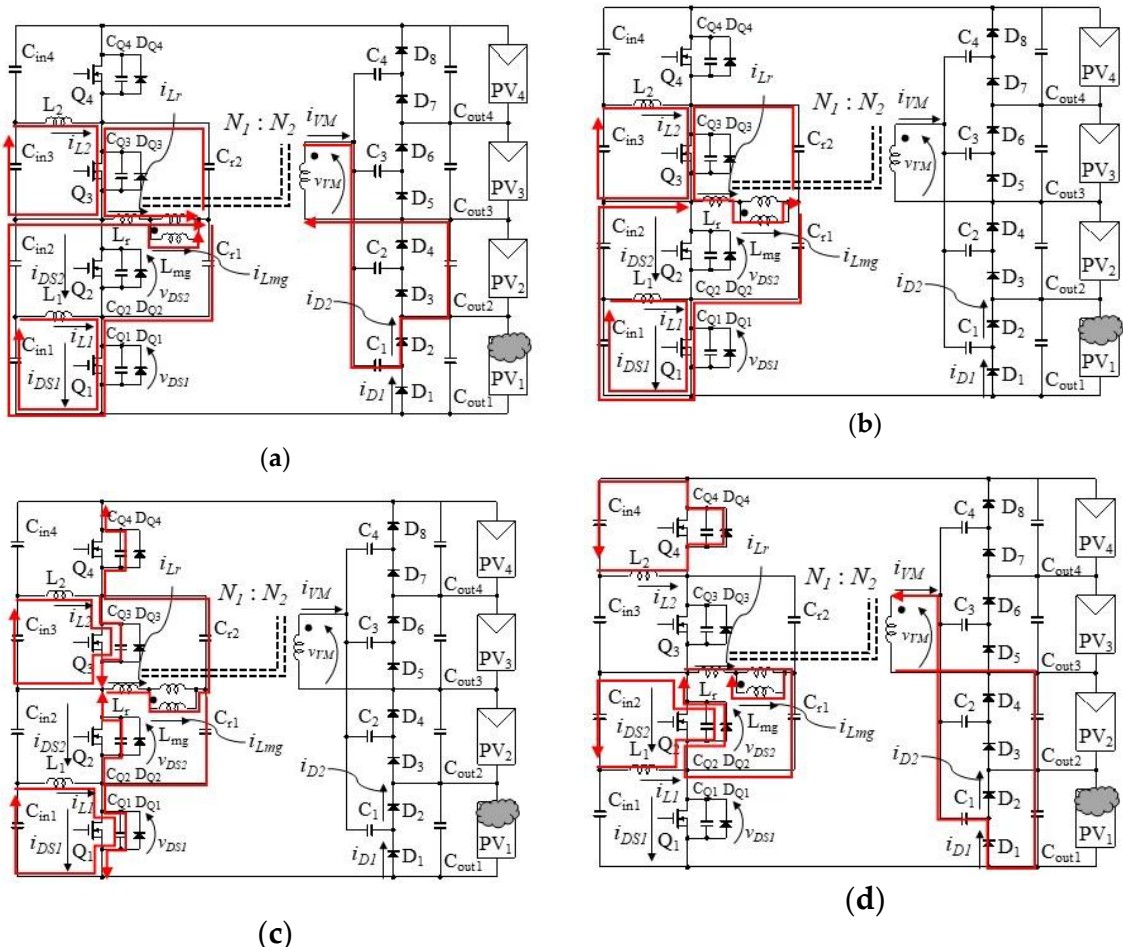

**Figure 9.** Operation modes: (**a**) Mode 1; (**b**) Mode 2; (**c**) Mode 3; (**d**) Mode 4.

*3.2. Operation Boundary*

In order for Modes 2 and 3 to exist, half the resonant period must be shorter than half the switching period. To this end, the following operation condition needs to be fulfilled;

$$\frac{1}{2\pi \sqrt{L_r(C_{r1} + C_{r2})}} = f_r \geq f_s \geq f_0 = \frac{1}{2\pi \sqrt{(L_r + L_{mg})(C_{r1} + C_{r2})}} \tag{1}$$

where $f_s$, $f_r$, and $f_0$ are the switching frequency, higher resonant frequency, and lower resonant frequency, respectively.

### 3.3. Voltage Equalization Mechanism of Voltage Multiplier

The VM, shown in Figure 10, is essentially a single-input-multi-output converter that converts an input ac voltage $v_{VM}$ with a peak-to-peak value of $V_{pp}$ into a dc voltage with a magnitude of $V_{pp}$. In Figure 10, for example, when an ac voltage $v_{VM}$ with $V_{pp} = V_{PV}$ is inputted, the VM produces uniform output voltages of $V_{PV}$ across $C_{out1}$–$C_{out4}$. $C_1$–$C_4$ in the VM behave as ac coupling capacitors that block dc currents and allow ac components only to flow through them. Hence, by removing dc components, the VM can be transformed to an equivalent circuit shown on the right-hand side, in which all the PV panels are separated and grounded. Since all PV panels are equivalently connected in parallel, all the voltages of panels as well as $C_{out1}$–$C_{out4}$ are automatically equalized. Detailed operation analyses have been performed in past works [28].

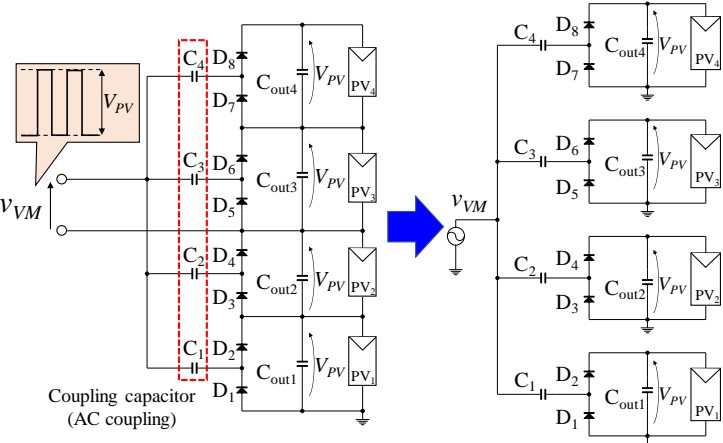

**Figure 10.** VM and its equivalent circuit.

### 3.4. Design Guideline for LLC Resonant Inverter

In general, a ratio of $L_{mg}$ to $L_r$ for ordinary LLC resonant converters is designed to be around 5–10 in order to obtain proper gain-frequency characteristics [34]; ratios vary depending on applications but do not exceed 10 in most applications. The proposed LLC resonant VM-based DPP converter, on the other hand, operates at a fixed switching frequency with a fixed duty cycle. Since ordinary PV panels are generally installed so that partial shading does not occur as far as possible, the proposed DPP converter should be designed to minimize the loss under unshaded conditions, which correspond to no-load conditions for ordinary LLC resonant converters. A small value in $L_{mg}$ leads to an increased loss due to a large $i_{Lmg}$ under no-load conditions. Hence, to reduce the loss originating from $i_{Lmg}$, $L_{mg}$ in the proposed DPP converter should be designed to be large within the range that $i_{Lmg}$ can completely charge and discharge $C_Q$ of MOSFETs in order to achieve the ZVS operations.

Since the ratio of $L_r$ to $L_{mg}$ of the LLC resonant tank (approximately 150 in the prototype, see Table 1) is rather larger than that of ordinary LLC resonant converter (around 5–10), the LLC resonant tank in the proposed DPP converter operates more like a series-resonant tank. Therefore, similar to traditional series-resonant tanks, the gain of the LLC resonant tank in the proposed DPP converter is unity at the higher resonant frequency $f_r$. It suggests the turns ratio of the transformer can be determined by assuming the unity gain of the LLC resonant tank.

**Table 1.** Circuit components used for prototype.

| Component | Value |
|---|---|
| $Q_1$–$Q_4$ | IRF644PSBF, $V_{DS}$ = 250 V, $R_{on}$ = 0.28 $\Omega$, $C_Q$ = 330 pF |
| $C_{in1}$–$C_{in4}$ | MLCC, 40 $\mu$F (= 10 $\mu$F × 4), 100 V |
| $C_{r1}$, $C_{r2}$ | Film Capacitor, 400 nF (= 100 nF × 4), 100 V |
| $L_1$, $L_2$ | 470 $\mu$H |
| $C_1$–$C_4$ | MLCC, 20 $\mu$F (= 10 $\mu$F × 2), 100 V |
| $C_{out1}$–$C_{out4}$ | MLCC, 50 $\mu$F (= 10 $\mu$F × 5), 50 V |
| $D_1$–$D_8$ | Schottky Diode, SBRT20M60SP5, $V_D$ = 0.57 V |
| Transformer | $N_1$:$N_2$ = 13:7, $L_r$ = 1.18 $\mu$H, $L_{mg}$ = 175 $\mu$H, $R_{Tp}$ = 1.77 $\Omega$, $R_{Ts}$ = 1.77 $\Omega$ |

The voltages of $C_{r1}$ and $C_{r2}$ are alternately applied to the transformer primary winding, and $V_{pp}$ is equal to $V_{string}$/2. As mentioned in Section 3.3, the voltages of the VM's outputs (i.e., $C_{out1}$–$C_{out4}$) are equal to the peak-to-peak value of the VM's input voltage $v_{VM}$ (or the voltage of the secondary winding), yielding the following equation:

$$\frac{N_2}{N_1} \frac{V_{string}}{2} = V_{PV} \tag{2}$$

where $V_{PV}$ is the voltage of the panel that receives an equalization current from the DPP converter. Assuming all voltages are ideally equalized to be $V_{PV}$, substituting $V_{string} = 4V_{PV}$ into (2) produces:

$$\frac{N_2}{N_1} = \frac{1}{2} \tag{3}$$

This equation corresponds to an ideal condition. In practical cases, however, considering forward voltage drops of diodes and voltage drops in resistive components in switches and capacitors, the turns ratio should be determined so that $2N_2$ is slightly larger than $N_1$ (i.e., $2N_2 > N_1$).

*3.5. Peak Current of Inductor*

Since the string voltage $V_{string}$ is divided by the VD, the voltages applied to the inductors in the VD can be reduced, contributing to reduced peak inductor currents. The peak current of the inductors, $I_{L.peak}$, can be yielded as:

$$I_{L.peak} = \frac{1}{2} \frac{V_{Cin} T_s}{2L} \tag{4}$$

where $V_{Cin}$ is the voltage of $C_{in1}$–$C_{in4}$, $T_s$ (= $1/f_s$) is the switching period, and $L$ is the inductance. Applying the experimental conditions ($V_{string}$ = 142.5 V, $V_{Cin}$ = 35.6 V, $T_s$ = 7.15 $\mu$s, and $L$ = 470 $\mu$H), $I_{L.peak}$ is calculated to be 136 mA. Although $V_{string}$ is 142.5 V, $I_{L.peak}$ can be as low as 136 mA with $L$ = 470 $\mu$H thanks to the zero average currents of the inductors.

## 4. Experimental Results

*4.1. Prototype*

Previous works reported that DPP converters capable of processing 30% of panel power are sufficient to preclude loss in annual energy yield in most cases [35,36]. As long as panels are installed in a proper location, severe shading conditions rarely occur. We assumed that two out of four panels are shaded at the same time and their short circuit currents are 30% less than those of unshaded panels in the worst shading condition. Accordingly, for a PV string consisting of four 200 W panels connected in series, each with a short-circuit current of 5.0 A and open-circuit voltage of 45 V, a prototype capable of supplying 60 W for each shaded substring was designed and built as shown in Figure 11. Component values used for the prototype are listed in Table 1. The maximum power rating of the prototype was 120 W. Except for $C_{r1}$ and $C_{r2}$, MLCCs were employed as capacitors in the prototype.

The higher resonant frequency $f_r$ was 164 kHz, and the prototype was operated at $f_s$ = 140 kHz with a dead time period of 0.4 µs.

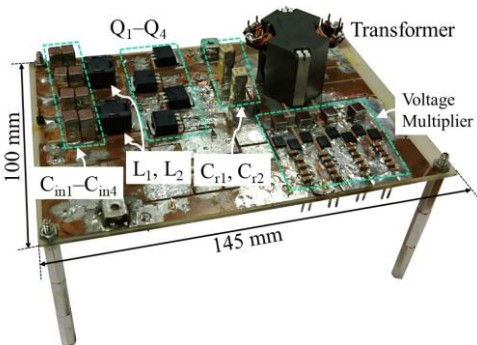

**Figure 11.** Prototype capable of supplying 60 W for each shaded panel in PV string consisting of four panels connected in series.

### 4.2. Fundamental Characteristics of DPP Converter

Power conversion efficiencies and output characteristics of the prototype were measured using the experimental setup shown in Figure 12. The input and output ports of the prototype were separated in order to measure power conversion efficiencies and output characteristics. All PV panels were removed, and the prototype was powered by an external power supply $V_{in}$. A variable resistor $R_{var}$ was connected to the output of the VM through the selective tap to emulate current flow paths under partial shading conditions. Selecting tap X, for example, can emulate the current flow paths under the case that $PV_1$ is partially shaded. With the tap Y selected, the prototype behaves as if $PV_1$ and $PV_2$ are uniformly shaded. $V_{in}$ was set to be 142.5 V, which corresponded to an MPP voltage of the string used for the experiment.

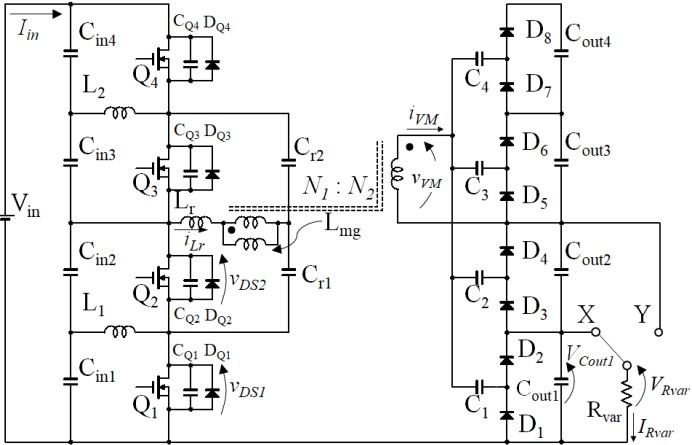

**Figure 12.** Experimental setup for characteristic measurement.

Measured key operation waveforms at $I_{Rvar}$ = 1.5 A with the tap X selected are shown in Figure 13. Peak voltages of $v_{DS1}$ and $v_{DS2}$ were about 71.3 V, nearly half the input voltage $V_{in}$ of 142.5 V, demonstrating that the input voltage was properly divided by the VD. Measured waveforms of $v_{GS1}$, $v_{GS2}$, $v_{DS1}$, and $v_{DS2}$ were in good agreement with theoretical ones (see Figure 8), demonstrating ZVS turn-on and turn-off operations.

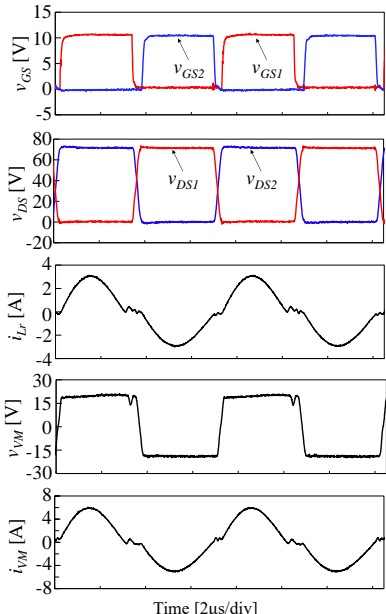

**Figure 13.** Measured key operation waveforms at $I_{Rvar}$ = 1.5 A with tap X selected.

Measured power conversion efficiencies and output characteristics are shown in Figure 14. $V_{Cout1}$ on the horizontal axis corresponds to the voltage of $C_{out1}$ or the voltage of the shaded panel in the practical case. Voltages of $C_{out1}$–$C_{out4}$ were nearly uniform thanks to the voltage equalization performance of the VM. The measured efficiencies were mostly higher than 90%. $V_{Cout1}$ linearly declined as $I_{Rvar}$ increased, and this tendency was more explicit when the tap Y was selected. It was because, with the tap Y selected, the input current of the prototype was twice that with the tap X selected, causing larger voltage drops in the VD on the transformer primary side.

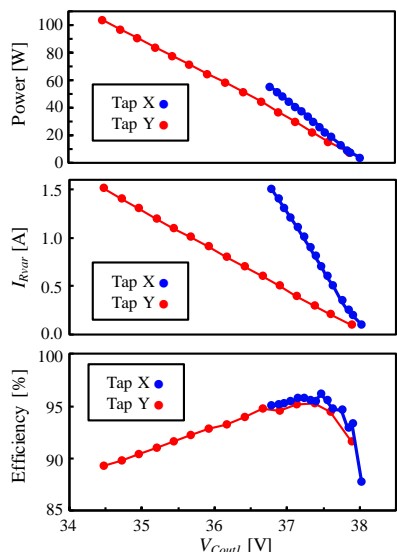

**Figure 14.** Measured power conversion efficiencies and output characteristics.

### 4.3. Equalization Test in Laboratory

Instead of actual PV panels, solar array simulators (E4361A, Keysight Technologies) were used to emulate a partial shading condition. Individual panel characteristics used for the laboratory experiment are shown in Figure 15a. PV$_3$ was a partially shaded panel, and its short-circuit current was 30% less than the others. The ideal maximum power of the string under this partial shading condition was

623 W. An electronic load was connected to the load port (see the port denoted as Load+ and Load−
in Figure 5a) in order to sweep the string characteristics. A string characteristic without the DPP
converter (i.e., with traditional bypass diodes) was also measured as a reference.

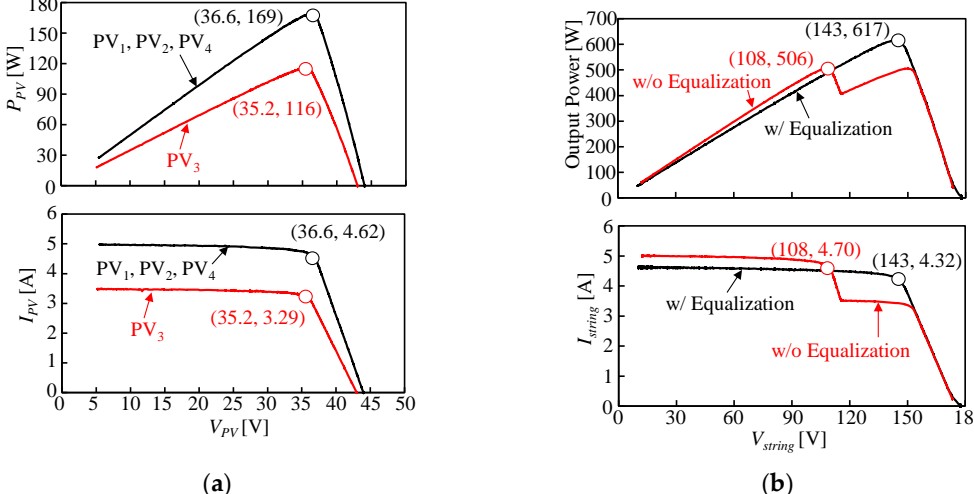

**Figure 15.** Experimental results of laboratory testing. (**a**) Individual panel characteristics used for
experiment; (**b**) Measured string characteristics with/without proposed DPP converter.

Measured string characteristics with/without the proposed DPP converter are compared in
Figure 15b. Two MPPs, including a local and global MPPs, were observed on the *P–V* characteristic in
the case without the DPP converter (i.e., with bypass diodes), and the maximum power was 506 W at
$V_{string}$ = 108 V. With the DPP converter, on the other hand, the local MPP disappeared, and the maximum
power increased to as high as 617 W at $V_{string}$ = 143 V, corresponding to 21.9%. The overall efficiency,
which is defined as the ratio of the extracted power to the ideal maximum power, was determined
to be 99.0% (= 617/623). The proposed DPP converter successfully increased the energy yield and
demonstrated its efficacy.

An image of power redistribution in laboratory testing is illustrated in Figure 16. Assume all the
panel voltages were uniform as 35.2 V. Since the current mismatch between the shaded and unshaded
panels was 1.33 A, the DPP converter supplied 46.8 W to the shaded panel of $PV_3$ for equalization.
Due to the support by the DPP converter, the string behaved as if it produced 670 W. However, since
53 W out of 670 W was inputted to the DPP converter, the extracted power at the load port was 617 W.
The power processed by the DPP converter under this partial shading condition was merely 53 W
and was rather smaller than the string power. This small processed power allowed the 99% overall
efficiency, even though the measured efficiency of the DPP converter was around 90% (see Figure 14).

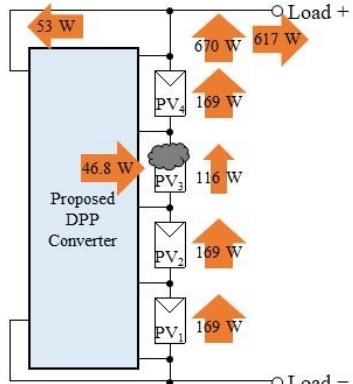

**Figure 16.** Power redistribution image in laboratory testing.

### 4.4. Field Testing

Field testing using an actual string consisting of four PV panels connected in series was performed, as shown in Figure 17. $PV_3$ was covered with a plastic bag to emulate the partial shading condition. Measured individual panel characteristics are shown in Figure 18a. The irradiance was measured to be 591 W/m² by a pyranometer (ES-602, EKO). The short-circuit current of $PV_3$ was 16.7% less than the others. The ideal maximum power under this shading condition was 672 W. Similar to the laboratory testing, string characteristics with/without the DPP converter were swept using the electronic load.

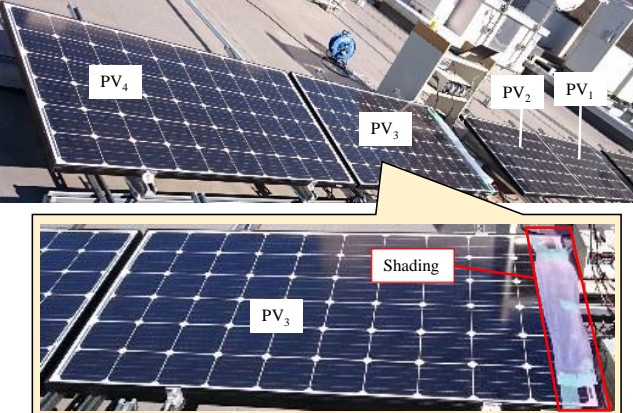

**Figure 17.** Experimental setup for field testing.

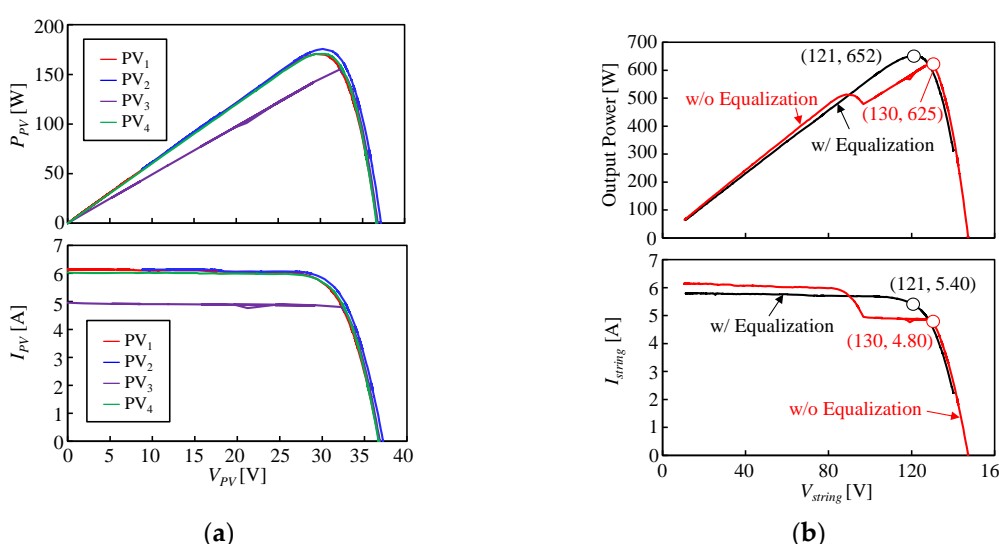

(**a**) (**b**)

**Figure 18.** Experimental results of field testing. (**a**) Individual panel characteristics used for experiment; (**b**) Measured string characteristics with/without proposed DPP converter.

Measured string characteristics are shown in Figure 18b. The string characteristic without the DPP converter (i.e., with traditional bypass diodes) exhibited two MPPs, and the maximum power at the global MPP was 625 W at $V_{string}$ = 130 V. With the DPP converter, the local MPP vanished, and the maximum power increased to 652 W at $V_{string}$ = 121 V, corresponding to the overall efficiency of 97.0% (= 652/672). These results demonstrate the improved energy yield of the actual PV string.

### 4.5. Extended DPP Converter for Eight Panels Connected in Series

A prototype of the extended topology for eight panels connected in series was also built and tested. Although the extended topology consists of more circuit components, component values used for the prototype were identical to those shown in Table 1. An equalization test was performed in the

laboratory using the solar array simulators. Individual panel characteristics used for the experiment are shown in Figure 19a. $PV_3$ and $PV_7$ were shaded panels, and their short-circuit current was 15% less than the others. The ideal maximum power of the string was 1294 W.

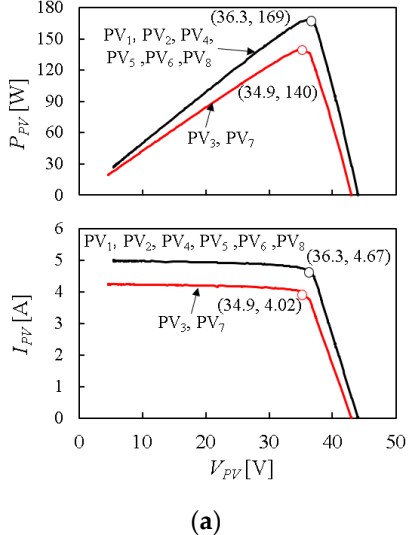

(**a**)

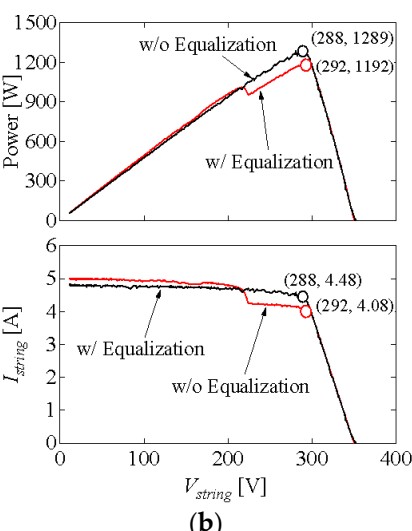

(**b**)

**Figure 19.** Experimental results of extended DPP converter. (**a**) Individual panel characteristics used for experiment; (**b**) Measured string characteristics with/without proposed DPP converter.

Measured string characteristics with/without the DPP converter are compared in Figure 19b. Two MPPs were observed when without the DPP converter, and the maximum power at the global MPP was 1192 W at $V_{string}$ = 292 V. With the DPP converter, on the other hand, the local MPP vanished, and the maximum power increased to 1289 W at $V_{string}$ = 288 V, corresponding to 8.2% improvement. The overall efficiency was as high as 99.6% (=1289/1294). The results demonstrated the extension concept of the proposed DPP converter.

## 5. Conclusions

This paper has proposed a DPP converter using an LLC resonant VM and VD. Voltage stresses of circuit components in conventional string-to-panel DPP converters are prone to soar as the string voltage increases. Meanwhile, the proposed DPP converter can reduce voltage stresses of switches and input smoothing capacitors thanks to the VD. The proposed DPP converter can automatically transfer power from unshaded panels to shaded ones, and therefore a feedback control loop can be eliminated, achieving simplified circuit implementation. The VD can be extended by adding switches and capacitors, and the voltage stresses can be further reduced by extending the VD.

Experimental verification tests using the prototype for four PV panels connected in series were performed emulating partial shading conditions in the laboratory and outdoor testing. The proposed DPP converter dramatically improved the energy yield and achieved the overall efficiency higher than 97%, demonstrating its performance. Furthermore, the prototype of the extended topology for eight panels connected in series was also built and tested, and its experimental results demonstrated the extension concept of the proposed DPP converter.

**Author Contributions:** Conceptualization, M.U.; methodology, M.U.; simulation analysis, T.N. and T.S.; validation, T.N.; writing—original draft preparation, M.U. and T.N.; writing—review and editing, M.U. and T.S.; supervision, M.U.

**Funding:** This research received no external funding.

**Conflicts of Interest:** The authors declare no conflict of interest.

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
