# Peer review of "LLC Resonant Voltage Multiplier-Based Differential Power Processing Converter Using Voltage Divider with Reduced Voltage Stress for Series-Connected Photovoltaic Panels under Partial Shading"

_electronics, doi:10.3390/electronics8101193_

Round 1

Reviewer 1 Report

The paper presents an improved solution to reduce the voltage stress for series-connected photovoltaic panels under partial shading. The main idea consist into a differential power processing based on the combination of an LLC resonant voltage multipliers and voltage divider using switched capacitor converters. Various kinds of differential power processing are presented and a comparison is performed.  The solution is validated by experimental verification tests in the laboratory and outdoor tests.

Manuscript’s strengths:

- Experimental verification tests using the prototype for four PV panels connected in series (emulating partial shading conditions in the laboratory and outdoor testing).

-  A clear and concise presentation of the results and improvements provided by the proposed solution.

Manuscript’s weaknesses:

- Testing and validation of the solution only for a small number of photovoltaic panels (small PV string).

Major recommendations for the improvement of the manuscript:

- The experimental tests are performed only for four PV panels connected in series were. Is the solution technically viable for a large string of photovoltaic panels? I think the case of the largest solar photovoltaic site should be analyzed. Also, economically speaking, what are the additional costs?

Minor recommendations for the improvement of the manuscript:

- The paper states that “this automatic power redistribution allows eliminating feedback control loop”.  Can you explain more detail this statement?

- The conclusions can be extended by adding additional details regarding the proposed solution.

- Regarding the English language, a few minor spell checks are required.

Author Response

We are very grateful to you for your thoughtful and helpful review of the manuscript. Your comments and suggestions have been incorporated as appropriate into the revised manuscript. The revised parts are highlighted with green in the revised manuscript. Please see the attached file.

Reviewer 2 Report

Interesting and well written contribution. The authors should emphasize the actual novelty of their work and address different points before being suitable for publication:

-Line 35: the authors are referring to the interest of the solution concerning the field of plug-in hybrid vehicles equipped with solar panels. This seems to be a very specific example, and probably not the most common one. Why just plug-in hybrids and not full EVs or just IC cars with a solar panel to cool down the passenger compartment?

-Line 42: the authors affirm that partial shading in PV “might confuse MPPT algorithms”. This statement looks like hypothesis, is this a real problem not solved by tracking algorithms?

-Line 63: In the adjacent panel-to-panel architecture the authors conclude that the system becomes complex. However, those converters are smaller, simpler and the extension to a larger number of series PV is sample since it is a modular solution. In the proposed solution the input voltage increases in proportion to the number of PVs, the converter becomes larger and more complex needing also a specific design for each configuration. What is the advantage?

- Line 69: The authors state that “isolated DPP” requires of a large transformer which takes a room in the board and it is expensive for each converter. However, the proposed DPP solution is a hybrid solution in between the “isolated DPP” and the “String-to-Panel DPP”. The proposed converter also requires of a transformer, despite being a single transformer for all the converter it is necessary to dimension it to provide full power (200 W in this proposal) in comparison with 60 W per panel with the other alternative.

- Figure 3. Please replace Vin by Vstring.

- Line 84: The topology has been previously presented in reference 27. This reviewer understands that the novelty here is to implement the multilevel LLC to reduce the voltage of the transistors. Please emphasize clearly the differences of this manuscript with reference [27].

-Why don't just use large resistors (MEG-Ohms) with low impact on the efficiency as a balancing technique for the input voltage instead of capacitors?

-Line 86: VD is not an SCC converter (it is not operating like that: there is not a net power transfer in any direction). As the authors wrote, the average inductor current is zero. Furthermore, the inductor is large to avoid peaks and large RMS, which is more important for losses, not the average current as stated several times in the paper.

-Line 125: Do you mean VD instead of VM?

-Line 143: “another prominent feature of the proposed DPP converter is good scalability”. I would say that this is a solution that can be scaled but “good” is a too optimistic attribute. Adjacent converters are more easily scaled. For example to go from n=4 to n=6, in this solution, another transformer is required.

-Line 195: “small inductors with a small current rating”. Average current is zero, but not peak or RMS current. Furthermore, the inductors are relatively large (470uH) and the core should be relatively large to avoid saturation. Please comment

-Line 218: ratio Lmg to Lr depends on the input and output range of the application. Please justify further the selection of a 5 ratio.

-The selection of the LLC operation frequency is not discussed. The converter is operated close to its resonant frequency but slightly lower, applying, therefore, a gain slightly over 1. Is this to compensate for the diodes voltage drop

-Why capacitors Cin1 to Cin4 are not the same as Cout1 a Cout4 in table 1?. From the analysis, the maximum voltage should be 45 V according to the panels.

-What is the maximum power that the converter can supply in the worst case of shading? Are the efficiency values at 60 W the most frequent case?

-Line 306. How the irradiance was determined?

-Line 315: the efficiency is determined by dividing 652/772 W, where does  772W come from?

Author Response

(The authors gave the same response as above.)

Round 2

Reviewer 2 Report

The authors have addressed most of my previous concerns in a satisfactory way. However, some points remain to be further clarified before the publication of this manuscript.

Comment #2: I still considered that the example provided for the application (solar roofs of electric vehicles and hybrid vehicles) is too specific; providing specific branding in the manuscripts worsens the problem. The authors answered "To the best of our knowledge, no IC cars have embedded solar roofs with a curved surface—car owners may add rigid or flexible solar panels by themselves. Nowadays, some automotive companies are selling or developing EVs and HEVs having embedded solar roofs with a curved surface. However, EVs and HEVs with solar roofs are not prevalent". For the knowledge of the authors: my 17 years old Volkswagen Passat has a PV panel embedded in the solar roof to cool down the passenger compartment when it is parked under the sun without using the main battery... that was part of the optional equipment at that time. It may not be a flexible PV but the idea was already there.
Please provide more examples of applications (and remove branding), otherwise, you should rewrite abstract and title to specify that you are targeting the specific problem of solar roofs in vehicles.

I do not agree with the statement that the proposed VD reduces the voltage stresses of switches. The proposed VD is a voltage balancing technique for the series capacitors but has no impact on the working principles of the LLC or the stress on the devices. The authors have not provided proof to this statement, e.g.: waveforms comparing the blocking voltages on the LLC devices. This concern must be further corroborated by simulation of the proposed topology alternatives or supported by previous results.

Comment #8: The reviewer does not understand the statement on the switching speed of the switches and its influence on the voltage of the stacked capacitors at the input of the converter. The authors have not provided an example waveforms or explanation to the stated immediate change in voltages in the stacked capacitors.
In my understanding, a common approach for the voltage balancing of capacitors is placing resistors accounting for a multiple of the leakage current of the capacitors themselves. For the ceramic capacitors, a reasonable value would be 1MOhm in parallel, which may consume in the order of 2 mW per resistor/capacitor at the maximum of 45 V in the example provided. In my opinion, the conduction and core losses of the additional inductors in the authors’ proposed solution may likely surpass those values.

Author Response

We are very grateful to you for your thoughtful and helpful review of the manuscript. Your comments and suggestions have been incorporated as appropriate into the revised manuscript.
